# Epileptic seizures diagnosis and prognosis from EEG signals using heterogeneous graph neural network

Areej Alasiry[1], Gabriel Avelino Sampedro[2], Ahmad Almadhor[3], Roben A. Juanatas[4], Shtwai Alsubai[5] and Vincent Karovic[6]

[1] College of Computer Science, King Khalid University, Abha, Saudi Arabia
[2] Department of Computer Science, University of the Philippines Diliman, Quezon City, Philippines
[3] Department of Computer Engineering and Networks, College of Computer and Information Sciences, Al Jouf University, Sakaka, Saudi Arabia
[4] College of Computing and Information Technologies, National University, Manila, Philippines
[5] College of Computer Engineering and Sciences, Prince Sattam bin Abdulaziz University, Alkharj, Saudi Arabia
[6] Faculty of Management, Comenius University in Bratislava, Bratislava, Slovakia



Corresponding authors
Gabriel Avelino Sampedro,
garsampedro@ieee.org
Vincent Karovic,
vincent.karovic6@fm.uniba.sk

## ABSTRACT

Epilepsy, often associated with neurodegenerative disorders following brain strokes, manifests as abnormal electrical activity bursts in the cerebral cortex, disrupting regular brain function. Electroencephalogram (EEG) recordings capture these distinctive brain signals, offering crucial insights into seizure detection and management. This study presents a novel approach leveraging a graph neural network (GNN) model with a heterogeneous graph representation to detect epileptic seizures from EEG data. Utilizing the well-established CHB-MIT EEG dataset for training and evaluation, the proposed method includes preprocessing steps such as signal segmentation, resampling, label encoding, normalization, and exploratory data analysis. We employed a standard train-test split with stratified sampling to ensure class distribution and reduce bias. Experimental comparisons with long short-term memory (LSTM) and recurrent neural network (RNN) models highlight the GNN's superior performance, achieving a classification accuracy of 98.0% and demonstrating incremental improvements in precision and F1-score. These findings emphasize the efficacy of GNN in capturing spatial and temporal dependencies within EEG data, surpassing conventional deep learning techniques. Furthermore, the study highlights the model's interpretability, which is essential for clinical decision-making. By advancing EEG-based seizure prediction methods, this research offers a robust framework for enhancing patient outcomes in epilepsy management while addressing the limitations of existing approaches.

## INTRODUCTION

Epilepsy is a chronic brain disease that causes epileptic seizures, abnormal behaviour, and loss of feeling or consciousness due to frequent and unpredictable interruptions in normal

brain function (*Duncan et al., 2024*). A chronic condition that has significant social and health implications is epilepsy. Over 50 million people worldwide suffer from the neurological disorder known as epilepsy. Anti-epilepsy drugs used to control this disease only work for 70% of people. However, about 30% of patients do not respond well to these therapies and need surgery (*WebMD Editorial Contributors, 2023*; *Shoeb, 2009*). According to reports, there is a severe lack of neurologists and the neurological assistance required of this staff, which can have a major impact on patients' timely access to treatment (*Majersik et al., 2021*). Thus, automatic seizure recognition is essential to help neurologists and other medical professionals diagnose patients more quickly and prescribe necessary therapies if needed.

The prevalence of epilepsy has increased among Alzheimer's disease (AD) patients. AD can increase the risk of seizures since it is a progressive disease starting with mild memory loss and possibly leading to loss of the ability to respond to the environment (*Zuo et al., 2024*; *Lei et al., 2023*; *Zuo et al., 2021*). Inappropriate electrical eruptions cause brain seizures in a cluster of brain cells and can happen in different brain parts. These attacks can happen anytime, like a few times a year or several times a day. It can vary from brief attention span gaps to severe and persistent convulsions. Short bursts of uncontrollable movement, known as recurrent seizures, can impact the whole body or just one part of the body. It results in diminished awareness and disruption of the function of the bladder and bowels (*Pandey et al., 2023*). The three main categories of status epilepticus are ictal, preictal, and interictal (*Fisher, Scharfman & DeCurtis, 2014*). Many patients still experience epileptic seizures and are not under complete control of their condition, even with the various therapies and interventions available to help manage the diseases.

People with epileptic seizures may endure serious psychological stress due to shame and loss of adequate social position (*Gupta, Bhattacharyya & Pachori, 2020*; *Bhattacharya, Baweja & Karri, 2022*). Customized innovations and artificial intelligence (AI) systems that use signals from the body, particularly customized for identifying seizure activity, have been established in response to the significant adverse effects epileptic seizures have on the standard lives of those who suffer them. Considering everything above, diagnosing these neurological disorders as soon as possible is vital. The procedures utilized for epileptic seizure identification and prediction include neurological evaluations (*Loring et al., 2011*), blood tests (*Tomkins et al., 2008*), and neuroimaging modalities (*Khodatars et al., 2021*). A neurological examination is a diagnostic method for assessing a person's physical abilities, mental capacity, and behaviour. Blood tests are used to look for indications of disease, genetic disorders, and further seizure signs. Neuropsychological examinations assess a person's thinking, speaking, and remembering. The test results help the doctor determine where seizures impact parts of the body.

Epileptic seizure identification using neuroimaging modalities necessitates extensive recording data for specialized clinicians to make informed decisions. However, doctors who analyze large amounts of neuroimaging data often end up with inaccurate diagnoses and prognoses of epileptic seizures. Eye strain from evaluating various functional or structural imaging modalities can contribute to this. Misdiagnosis can also result from

various sounds in neuroimaging modalities. To overcome these challenges, neuroimaging modalities have been used to develop computer-aided diagnosis systems (CADS) for epileptic seizure identification, which will help physicians make accurate diagnoses.

AI is gaining popularity in various medical fields due to its ability to analyze large volumes of medical data, provide insights for diagnosis and treatment, assist in medical imaging interpretation, predict patient outcomes, personalize treatment plans, and improve overall healthcare efficiency and quality (_Noor et al., 2020_; _Murphy et al., 2021_). AI is a collection of algorithms that can simulate certain human abilities, like learning and making predictions based on that learning (_Viner et al., 2020_). Deep learning (DL) and machine learning (ML) are subcategories of AI. ML is the collection of algorithms and methods that let computers process data without following explicit instructions from a computer program. Many-layered artificial neural networks (ANNs) are the basis of DL, a subset of ML methods used in recent research. Numerous studies on AI approaches for epileptic seizure identification have been carried out thus far.

Despite the advances in conventional ML and DL techniques, their ability to effectively model the spatial and temporal dependencies inherent in EEG signals remains a significant limitation. Graph neural networks (GNNs) offer a promising alternative, leveraging heterogeneous graph structures to capture intricate relationships between EEG channels. This makes GNNs particularly suited for the task of epileptic seizure detection, where accurate identification of seizure events is crucial for patient safety and clinical management. This research proposed an approach that used the CHB-MIT dataset gathered using an EEG signal and employed the GNN algorithm to classify seizure occurrence. By addressing the limitations of existing approaches and leveraging the unique capabilities of GNNs, this study aims to provide a robust, interpretable framework for EEG-based seizure detection, ultimately contributing to improved clinical decision-making and patient outcomes.

## Research contribution

This study's main contributions are explained in more detail in the list format below.

- This study proposes a heterogeneous GNN model for epilepsy prediction using the CHB-MIT dataset that extends the capabilities of Conventional GNNs by accommodating various types of nodes and edges in the graph.
- This study performs exploratory data analysis and utilizes data preprocessing approaches, including signal segmentation, resampling, label encoder (one hot-encoding) and normalization (Z-score Normalization). Further, it employs a standard train-test split with stratified sampling to ensure class distribution and reduce bias.
- The research article uses several evaluation measurements such as accuracy, precision, recall and F1-score to analyze the model's efficacy for identifying seizures and no-seizures. The results demonstrate that the proposed GNN approach correctly predicts epilepsy and surpasses the performance of other DL models.

### Research organization

The study technique for classifying epilepsy is described as follows: A survey of the literature on DL and ML methodologies for classifying epilepsy detection is presented in "Related Work". "Proposed Approach" provides the research approach for the proposed work, which utilizes the CHB_MIT dataset, a deep learning model, and data preprocessing. The conclusions and findings are discussed and explained in "Experimental Result and Analysis". The work's conclusion and recommendations for future direction are contained in "Conclusion".

## RELATED WORK

*Lemoine et al. (2023)* created an approach based on automatic EEG processing to predict seizure recurrences in individuals receiving recurring EEG for one year. A retrospective selection was made between a temporally shifted cohort of 261 patients, the testing set, and a sequential cohort of 517 individuals receiving conventional EEG at their institution, the training set. After that, they created an automated processing pipeline to take the EEG data and extract both linear and non-linear features. Using multichannel EEG segments, they developed machine-learning algorithms to predict the recurrence of seizures after a year. They verified the results of the testing set and assessed how IEDs and clinical variables affected the performance. After applying the experiment, the ROC-AUC for seizure occurrence after EEG was 0.63. This research employs ML approaches to differentiate between individuals with idiopathic generalized Epilepsy and healthy controls based on interictal electroencephalogram recordings (*Escobar-Ipuz et al., 2023*). The presented work uses a scalp EEG scan to predict whether patients have idiopathic widespread Epilepsy. Moreover, the primary focus of this work is the extreme gradient boosting (XGB) technique utilized for scalp EEG. This XGB attempts to find signals from brain recordings of electroencephalograms that would permit the detection of IGE with elevated precision and separate IGE patients from normal commands, giving physicians an extra instrument to help in their decision-making. The proposed XGB approach, out of all the ML techniques used, produces a superior prediction of the unique attributes in EEG signals from IGE patients. XGB outperformed the k-nearest neighbors approach by 6.26% and outperformed the decision tree (9.71%), support vector machine (10.61%), and Gaussian naïve Bayes (11.83%) in terms of accuracy. Furthermore, out of all the methods examined, the proposed XGB technique had the highest performance in terms of AUC with a value of 98% and accuracy with a value of 98.13%.

*Singh & Lobiyal (2023)* developed a hybrid model for epileptic seizure prediction that consists of an LSTM and a deep convolution network (ResNet50) using EEG data. The proposed hybrid model is trained on spectrogram images to extract features and classify data. They examined the CHB-MIT scalp EEG dataset. Validation is performed to assess the efficacy of the proposed model for each preictal phase of 5, 15, and 30 min. According to the experimental findings, a 5-min preictal length was the ideal performance for the proposed model. The performed experiment results showed the proposed model achieved good performance in terms of F1-score with a value of 93%, a false positive rate (FPR) with a value of 0.055, an accuracy with a value of 94.5%, and a sensitivity with the value of

93.7%. In article *Miron et al. (2023)*, a supervised ML technique was employed to analyze EEG data obtained from a minimally intrusive procedure using foramen ovale (FO) and epidural peg electrodes to predict post-surgical seizure independence. Power-spectral EEG parameters were combined into a logistic regression (LR) model to predict seizure relief one year after surgery. The prediction model was compared to the outcome produced by clinical and scalp EEG factors after being validated *via* repeated 5-fold cross-validation. The study comprised 47 patients, of whom 26 had 1-year post-surgical seizure independence; the remaining 27 patients had peg-onset seizures, and 31 had FO. The AUC-ROC for post-surgical seizure relief prediction using electrophysiological characteristics was $0.74 \pm 0.23$ in patients with FO onset seizures. On the other hand, the AUC for the predictions derived from the clinical and scalp EEG components was $0.66 \pm 0.22$.

*Lu et al. (2023)* proposed a CBAM-3D CNN-LSTM model to predict epilepsy seizures. First, they perform a short-term Fourier transform (STFT) to preprocess EEG signals. Then, the preictal and interictal stage attributes were dragged from the preprocessed signals using the 3D CNN model. At last, for classification, BiLSTM is linked to 3D CNN. Lastly, the model incorporates CBAM. For the model to correctly extract interictal and preictal features, different considerations are provided to the data channel and space to drag essential facts. The proposed method was produced in the CHBMIT publicly available scalp EEG dataset. It attained a performance in terms of accuracy with a value of 97.95%, a sensitivity with a value of 98.40%, and a false alarm rate of 0.017 h1. Five DL algorithms based on intracranial EEG datasets are proposed in *Ouichka, Echtioui & Hamam (2022)* to automatically predict epileptic seizures. The convolutional neural network (CNN) model, the Fusion of Two CNNs, the Fusion of Three CNNs, the Fusion of Four CNNs, and transfer learning using ResNet50 are the foundations of the proposed approach. The experimental findings demonstrate that the 3-CNN and 4-CNN-based proposed procedures produced the greatest outcomes. Both of them attain 95% accuracy. *Nanthini et al. (2022)* offered an LSTM model with an EEG dataset to detect and predict seizure states. The dataset used, publicly available in Kaggle's comma-separated value release and housed in the UCI Database, is fed into the proposed model for validation. Using LSTM Networks, the proposed work has attained performance with a value of 99%.

*Bhattacharya, Baweja & Karri (2022)* proposed a seizure prediction model with a transformer model based on deep learning and Fourier transform feature extraction that consumes the potential components to automatically recognize the observant areas in EEG signals for efficient screening. This is done through an assortment of signal processing and DL algorithms. Utilizing the benchmark dataset, the proposed pipeline has improved sensitivity and FPR/h performance with a value of 98.46%, 94.83%, and 0.12439, 0. Epileptic seizures are presented in *Usman, Khalid & Bashir (2021)*. According to the proposed procedure, bandpass filtering removes noise from EEG data after undergoing empirical mode decomposition. Using artificial preictal segments produced by generative adversarial networks, the issue of class imbalance has been lessened. A three-layer modified CNN approach was created to retrieve automatic attributes from preprocessed EEG signals and combine them with manual characteristics to produce an extensive

attribute set. Following that, model-agnostic meta-learning is utilized to train a classifier using the feature set, which integrates the outcome of SVM, CNN, and LSTM. The proposed method has attained a good performance in terms of sensitivity and specificity with a value of 96.28% and 95.65%.

*Wu et al. (2022)* presented a technique using sequential variational mode decomposition (SVMD) and transformers for epileptic seizure prediction. A multidimensional version of SVMD is added for the time-frequency examination of multichannel signals. The proposed seizure prediction method involves using multivariate SVMD to break down the data into many modes on various time scales and then remove any unnecessary modes for preprocessing. Lastly, pre-trained bidirectional encoder representations (BERTs) receive the power spectrum of the denoised data as input for the prediction. The BERT may determine how data is connected to epileptic seizures in the timefrequency environment. On an intracranial EEG dataset, its prediction performance is fair, with an average sensitivity of 0.86 and an FPR of 0.18/h. *Jemal et al. (2022)* analyzed an interpretable DL algorithm for epileptic EEG-driven seizure prediction. This neural network is understandable because of its distinct architecture, which allows for the visualization and interpretation of its layers—the learned weights are obtained from signal analysis calculations such as frequency sub-band and spatial filters. As a result, since the derived features match those frequently employed for decoding EEG data, they are no longer abstract. The proposed approach performed better than previous techniques provided to predict seizures, utilizing the CHB-MIT dataset. The signals coming from streams in the brain area where the seizure starts contribute most substantially to the network predictions, and the first network layer filters align with clinically important frequency ranges. *Zambrana-Vinaroz et al. (2022)* proposed a seizure-predicting approach established on ear EEG, ECG, and PPG signals received utilizing an instrument operated in a fixed and outpatient context. People with epilepsy have been tested for this device in a clinical setting. Several predicting models that can classify the condition of the epileptic patient into regular, pre-seizure, and seizure have been constructed by analyzing this data and utilizing supervised ML algorithms. Following this, a Boosted Trees-based reduction model has been verified, yielding a 91.5% prediction accuracy and 85.4% sensitivity.

Table 1 provides the overview of the summary of the related work on epilepsy detection. According to the literature, this research needs to address some gaps. Firstly, the authors utilized machine learning approaches that are limited in performance (*Cui et al., 2018*; *Wang et al., 2019*). Further research extends by implementing deep learning classifiers in *Yao, Cheng & Zhang (2019)*, *Varnosfaderani et al. (2021)*, *Dissanayake et al. (2020)*, but are also limited in performance and apply to single deep learning classifiers. No research implements advanced versions of deep learning models such as a hybrid approach, heterogenous methods and proper preprocessing that can increase a model's performance. By addressing this limitation, we proposed a heterogeneous-based deep learning approach by utilizing numerous datasets from CHB-MIT to test the generalizability of our approach.

**Table 1 Summary of the related work.**

| Reference | Approach | Performance | Advantage | Limitation |
|---|---|---|---|---|
| Cui et al. (2018) | ELM | 75% | Computation time less | Low performance |
| Ibrahim et al. (2019) | Thresholds | 85.2% | Approach applicable in mobile apps for epilepsy patients | Low performance |
| Yao, Cheng & Zhang (2019) | RNN | 87.1% | Parameter Setting increase performance | Low performance |
| Wang et al. (2019) | RF | 84.00% | RF outperforms for the preictal state prediction | Low performance |
| Varnosfaderani et al. (2021) | LSTM | 85.1% | Highest AUC score of 0.92 and the lowest FPR of 0.147 | Low accuracy |
| Rasheed et al. (2021) | DCGAN | 88.21% | Low false prediction rate of 0.14/h | Low performance |
| Dissanayake et al. (2020) | CNN | 88.81% | Model outperforms for patient-independent seizure prediction | Low performance |

## PROPOSED APPROACH

The presented model approach used the CHB-MIT scalp EEG dataset from recorded EEG signals to detect epileptic seizures that precisely predict the seizures and no-seizure events. The proposed approach contains many phases: experimental dataset, exploratory data analysis and preprocessing, labelling, and normalization. Moreover, it presented model predictions employing DL classifiers. The presented model detail working is shown in Fig. 1.

In the first stage, the CHB-MIT dataset is collected from EEG signals, and seizure and no-seizure events are used to predict epilepsy disorder. Next, data preprocessing and analysis are needed for the collected data because they include missing values. At this step, duplicate data are eradicated, issues are resolved, and data is converted into a consistent structure. During this step, the data are encoded using a label encoder, and the data is normalized using normalization (standard scalar technique). Lastly, to detect epilepsy disorder, DL models (recurrent neural network (RNN) and LSTM) and proposed GNN are trained on this preprocessed dataset.

### Experimental dataset and preprocessing

The first step in any detection system is selecting inputs for classification. Brain activity, measurable by electroencephalogram (EEG), begins in childhood and continues throughout life, reflecting overall health. For epilepsy detection, we used EEG signals from the CHB-MIT dataset, which includes scalp EEG recordings from ten pediatric patients (ages 1.5 to 15) with uncontrollable seizures (John, 2010; Deepa & Ramesh, 2021). This dataset, widely used for real-time automated seizure detection, supports our goal of developing a generalized EEG seizure sensor independent of specific patients or channels. Data preprocessing is crucial for improving model precision and efficiency. This section implements data preprocessing steps, including exploratory data analysis (EDA) and data cleaning, data splitting, normalizing the data using normalization (Z-score normalization) approaches, and turning the categorical data into numerical values.

#### Signal segmentation and resampling

The EEG recordings were segmented into fixed-length epochs of 10 s to ensure uniform data segments. All signals were resampled at 256 Hz to maintain a consistent sampling rate

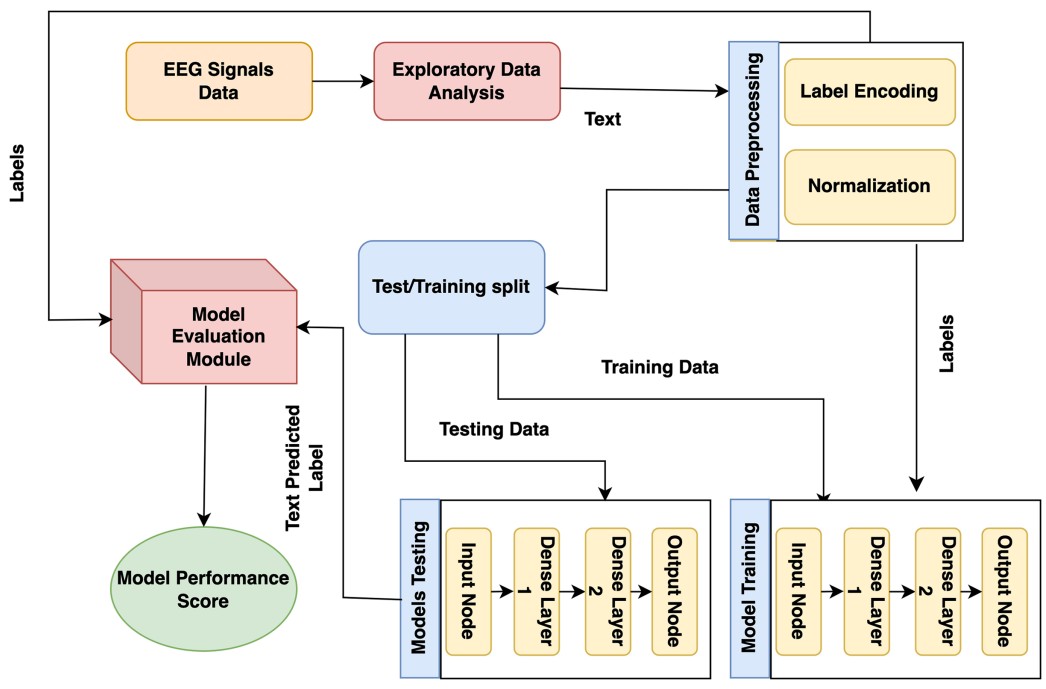

**Figure 1 Proposed architecture for epilepsy prediction.**

across the dataset. The noise was removed by applying a bandpass filter (0.5–50 Hz), and the signals were normalized to zero mean and unit variance. These preprocessing steps ensured that all data segments were consistent in length and quality and reduced noise for improved analysis.

### Label encoder

A label encoder is a preprocessing approach utilized in ML classifiers to convert categorical data, which consists of non-numeric labels, into a numerical format. Many ML classifiers utilized for different methodologies need numerical input, and label encoding is an approach that is mostly utilized to accomplish this transformation (*Takahashi et al., 2020*).

### Normalization

Normalization is a preprocessing technique in machine learning that scales and transforms the features of a dataset to bring them within a similar range. Normalization ensures that no single feature dominates the others, especially when features have different units or scales. This step is essential for some, depending on the size of the input characteristics. Two techniques are mostly used for normalization: Min-Max Scaling (MinMaxScaler) and Z-score normalization (Standard Scaler). In this research, we used Z-score normalization (Standard Scaler), standardizing the data to have a mean of 0 and a standard deviation of 1.

$$X_{normalized} = \frac{X - X_{mean}}{X_{std}}. \tag{1}$$

Equation (1) represents the normalization process applied to a variable $X$ to obtain its normalized counterpart $X\_normalized$. $X$ represents the original variable we want to normalize. It could be any numerical feature or dataset. $X\_mean$ denotes the mean (average) 5 value of the variable $X$. Subtracting the mean from each data point centres the distribution around 0, a common normalization step. $X\_std$ represents the standard deviation of the variable $X$. Dividing each data point by the standard deviation scales the data, ensuring a standard deviation of 1. Each data point $X$ is transformed into its normalized counterpart $X\_normalized$, with a mean of 0 and a standard deviation of 1. This normalization process ensures that variables with different scales and units can be compared consistently, making it easier to interpret and analyze the data.

## Recurrent neural networks

RNNs are DL classifiers that process data sequentially. RNNs differ from standard feedforward neural networks in that they feature connectivity that forms internal cycles, enabling the network to remember prior data. Because of this, RNNs are especially effective for sequence-related tasks like speech recognition, natural language processing, and time series assessment. The basic idea behind RNNs is to use information from previous time steps to influence the current prediction or output. This enables the network to capture temporal dependencies in the data. However, conventional RNNs have drawbacks. For example, they have trouble learning long-term relationships because of problems like vanishing gradients. This study used a simple RNN layer with a single recurrently connected hidden layer. The number of units in the RNN layer is 32, which determines the size of the hidden state and the activation function used in the RNN layer, rectified linear unit (ReLU). The dense layer is used as the output layer, and the following parameters include num_classes, and the activation function is softmax.

## Long short-term memory

LSTM was produced to successfully capture long-range relationships in data and solve the vanishing gradient issue that RNNs have. Applications combining time series data and natural language processing benefit greatly from the use of LSTMs (*Alsubai et al., 2022*). This study applied an LSTM model with two LSTM layers and two dropout layers. It then compiles the model with the sparse categorical cross-entropy loss function and the Adam optimizer. The model is trained using a custom training loop, where the gradients are computed and applied manually. The key parameters of an LSTM layer are units, return_sequences, dropout and recurrent_dropout. An LSTM model trains for 200 epochs with a batch size of 32.

## Graph neural network

GNN is a framework developed to proceed and illustrate data in graphs or networks. Graphs here comprise nodes, also called vertices, connected by edges and links. GNNs are applied to obtain information and derive understandings from data (*Scarselli et al., 2008*; *Zhou et al., 2020*). Figure 2 depicts the architecture of the GNN model. The GNN model in

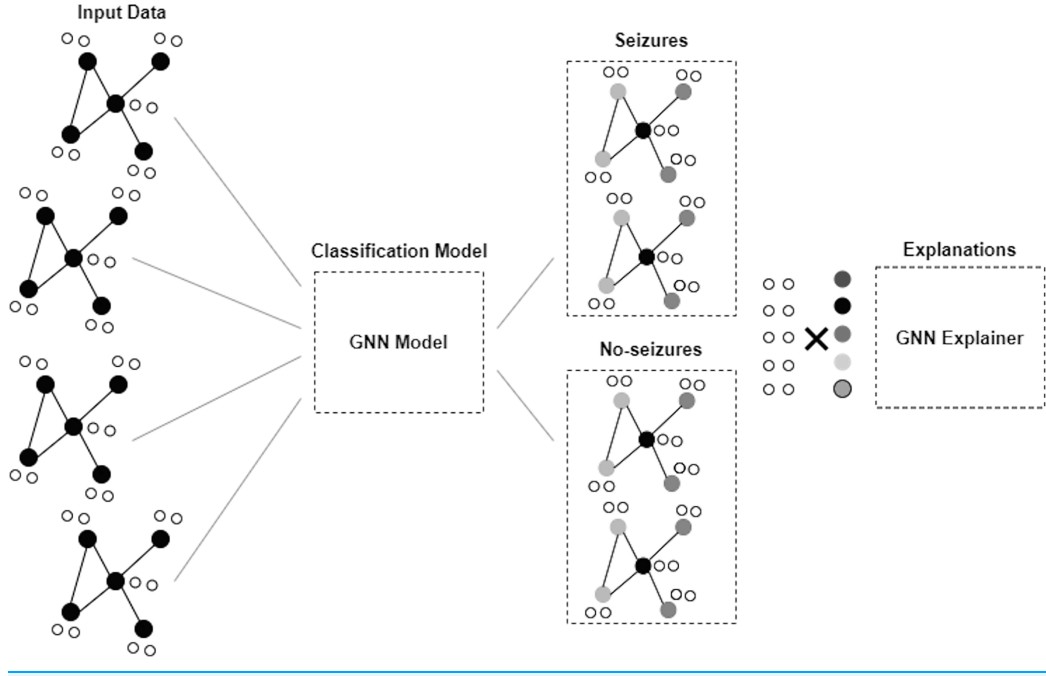

**Figure 2 Architecture of GNN model.**

this study consists of two dense layers. The first layer is a fully connected dense layer with 16 units. ReLU activation is chosen to introduce non-linearity and enable the model to learn complex patterns in the data. The second layer consists of num_classes units and softmax activation that are used to output the probability distribution over the classes for multi-class classification.

The loss function used is sparse categorical cross-entropy, and the optimizer is Adam. We used the sparse categorical cross-entropy loss function because it is well-suited for multi-class classification tasks where integers encode class labels. This choice helps the model optimize for the correct class predictions by minimizing the cross-entropy between the true and predicted distributions. The Adam optimizer was selected due to its adaptive learning rate properties, which provide efficient and robust training performance across a wide range of deep learning tasks. Adam combines the benefits of both AdaGrad and RMSProp, making it suitable for dealing with sparse gradients and noisy data. The number of epochs was set to 50, and the batch size was set to 32. These values were chosen based on preliminary experiments to balance training time and model performance. The epoch count ensures sufficient iterations for the model to converge, while the batch size balances computational efficiency and gradient estimate stability.

The proposed approach for epilepsy disease prediction on the CHB-MIT dataset is presented in Algorithm 1. Using CHB-MIT data as input, the system outputs predicted epilepsy disorders with seizure or no-seizure episodes. The three primary tasks of the algorithm are to assemble a dataset, train the model, and predict ratings. The main

| **Algorithm 1** Pseudo code of epilepsy disease detection. |
|---|

1: **Input:** CHB-MIT Dataset

2: **Output:** Epilepsy Disease Detection

3: $Data \leftarrow$ Assemble (CHB-MITDataset)

4: $D_A \leftarrow$ Exploratory Data Analysis (Data)

5: **function** $D_P \leftarrow$ Data Preprocessing (Data):

6:        Handling missing data and duplicate Data

7:        $D_{norm} \leftarrow$ Normalization ($D_P$)

8:        X, Y  Split($D_{norm}$ )

9: **Return** X (features), Y (labels)

10: **function** TrainingModels (X, Y)

11:        $D_s \leftarrow$ Split Dataset = training and test

12:        $D_L \leftarrow$ Utilized following Deep Learning Classifiers

13:        RNN() $\leftarrow$ Recurrent Neural Network ($D_s$)

14:        LSTM() $\leftarrow$ Long Short Term Memory ($D_s$)

15:        GNN() $\leftarrow$ Graphical Neural Network ($D_s$)

16: **return** *Trained Models* {Trained models are returned for prediction in this function}

17: **function** Prediction *(Trained Models)*

18: $E_m \leftarrow$ Evaluation Measurements

19: **return** $\leftarrow$ Results {This function returned result by evaluating different evaluation metrics}

function combines the three functions to assemble the entire prediction system. Using EEG signals, the Assemble() function takes pertinent data from patient records. Following data retrieval, analysis and preprocessing are performed to detect missing values, eliminate duplicate data, encode labels, and apply the normalization procedure to transform the data into a standard format. After completing these procedures, the dataset is thoroughly cleaned and ready to be input into the DL model to build a prediction model. The preprocessed dataset is finally returned by the functions X and Y. The TrainingModel() function creates a DL model from the preprocessed dataset. The used dataset is divided into 20% and 80% weightage for testing and training data. The function finally provided the trained model in return. The trained function utilizes the model to categorize seizures and non-seizures. After receiving the preprocessed dataset, the function outputs the predicted classes and the trained model as inputs.

## EXPERIMENTAL RESULT AND ANALYSIS

The research results and analysis of the classification are explained in this section. This section explains the deep learning model evaluation on the CHB-MIT dataset. The results show that the proposed model can effectively predict seizures while handling the subject variability in EEG signals.

## Evaluation measurements

The evaluation measures provided below are used to determine how effective the proposed model is. Among the measurements utilized to assess the classification problems are the accuracy provided in Eq. (2), the precision in Eq. (3), the recall in Eq. (4), and the F1-score in Eq. (5). A confusion matrix (CM) table can assess a classification model's performance by contrasting its expected and actual outputs. Machine learning is frequently used to assess a categorization model's efficacy. Each of the four quadrants of a CM—true positive (TP), false positive (FP), true negative (TN), and false negative (FN)—represents a potential result. The matrix columns illustrate the anticipated class labels, and the rows illustrate the actual ones. The matrix's main diagonal represents the successfully categorized examples, and the off-diagonal entries illustrate the incorrectly classified samples. These measurements aid in determining the model's advantages and disadvantages and can be applied to enhance the model's performance.

$$Accuracy = \frac{TP + TN}{TP + TN + FP + FN} \tag{2}$$

$$Precision = \frac{TP}{TP + FP} \tag{3}$$

$$Recall = \frac{TP}{TN + FN} \tag{4}$$

$$F1 - score = 2 \times \frac{Precision \times Recall}{Precision + Recall}. \tag{5}$$

The receiver operating characteristic (ROC) curve is a graphical representation used to evaluate the performance of binary classification systems. It plots the true positive rate (TPR) against the false positive rate (FPR) at various threshold settings, providing a comprehensive visualization of a model's ability to distinguish between positive and negative classes. A model with a ROC curve closer to the top-left corner indicates a higher discriminative ability. In contrast, the area under the curve (AUC) summarizes the overall performance: a value of 1 represents perfect accuracy, while 0.5 suggests no better performance than random guessing. This section provides a detailed discussion of the experiment outcomes.

The classification performance results for RNN, LSTM, and GNN models are shown in Table 2. GNN has the highest accuracy, 98.00%, for both the "No-Seizure" and "Seizures" classes, indicating that it makes correct predictions more often than RNN and LSTM. LSTM achieves an accuracy of 97.00%, while RNN achieves 98%. LSTM achieves testing F1-scores of 98.00% for the "Seizures" class and "No-Seizure" class. RNN achieves F1-scores of 98.00% for the "No-Seizure" and "Seizures" classes, respectively.

Table 3 presents the time required for training and testing predictions for the LSTM, RNN, and GNN models. Among the three models, the GNN model is the fastest, with a training time of 0.0081 s and a testing time of 0.0054 s, demonstrating its computational efficiency. The LSTM model takes the longest time for training, requiring 0.1731 s, while its testing time is slightly shorter at 0.126 s. The RNN model has a moderate training time of 0.029 s and a testing time of 0.0144 s. These results highlight that the GNN model

**Table 2 Classification reports for seizure prediction models.**

|  | Labels | Precision | Recall | F1-score | Support |
|---|---|---|---|---|---|
| RNN model (Train report) | Noseizures | 0.97 | 0.99 | 0.98 | 13,492 |
|  | Seizures | 0.99 | 0.97 | 0.98 | 13,541 |
|  | Accuracy | 0.98 |  |  | 27,033 |
| RNN model (Train report) | Noseizures | 0.97 | 0.99 | 0.98 | 3,404 |
|  | Seizures | 0.99 | 0.97 | 0.98 | 3,355 |
|  | Accuracy | 0.98 |  |  | 6,759 |
| LSTM model (Train report) | Noseizures | 0.95 | 0.99 | 0.97 | 13,492 |
|  | Seizures | 0.99 | 0.95 | 0.97 | 13,541 |
|  | Accuracy | 0.97 |  |  | 27,033 |
| LSTM model (Test report) | Noseizures | 0.96 | 0.99 | 0.98 | 3,404 |
|  | Seizures | 0.99 | 0.96 | 0.98 | 3,355 |
|  | Accuracy | 0.98 |  |  | 6,759 |
| GNN model (Train report) | Noseizures | 0.97 | 1.00 | 0.98 | 13,492 |
|  | Seizures | 1.00 | 0.97 | 0.98 | 13,541 |
|  | Accuracy | 0.98 |  |  | 27,033 |
| GNN model (Test report) | Noseizures | 0.98 | 0.99 | 0.98 | 3,404 |
|  | Seizures | 0.99 | 0.98 | 0.98 | 3,355 |
|  | Accuracy | 0.98 |  |  | 6,759 |

**Table 3 Time prediction for training and testing of models.**

| Model | Training time (s) | Testing time (s) |
|---|---|---|
| RNN model | 0.0429 | 0.0144 |
| LSTM model | 0.1731 | 0.1266 |
| GNN model | 0.0081 | 0.0054 |

significantly outperforms RNN and LSTM in terms of computational speed, making it more suitable for applications requiring real-time predictions.

Figure 3 depicts the graphical representation of the RNN model by evaluation training, testing loss, accuracy, CM, and ROC. The left Fig. 3A shows the training and validation loss. Validation loss starts from $0^{th}$ epoch with value 0.18%, and after some fluctuation increase and decrease, it decreases to 0.1% at $50^{th}$ epochs. The right graph in 3A displays the model's training and validation accuracy. Validation accuracy starts from $0^{th}$ epoch with a value around 0.95%. After some fluctuation increase and decrease, it stops at 0.98% at $50^{th}$ epoch. The left graph in Fig. 3B is a training confusion matrix that shows it performs better because the proposed strategy produces more continuous, better true positive (13,421) and negative (71) values and fewer false positive (365) and negative (13,175) results. The right graph in Fig. 3B is a testing confusion matrix. It also produces more continuous, better true positive (3,374) and negative (30) values and fewer false positive (89) and negative (3,266) results. Figure 3C displays the ROC curves for both

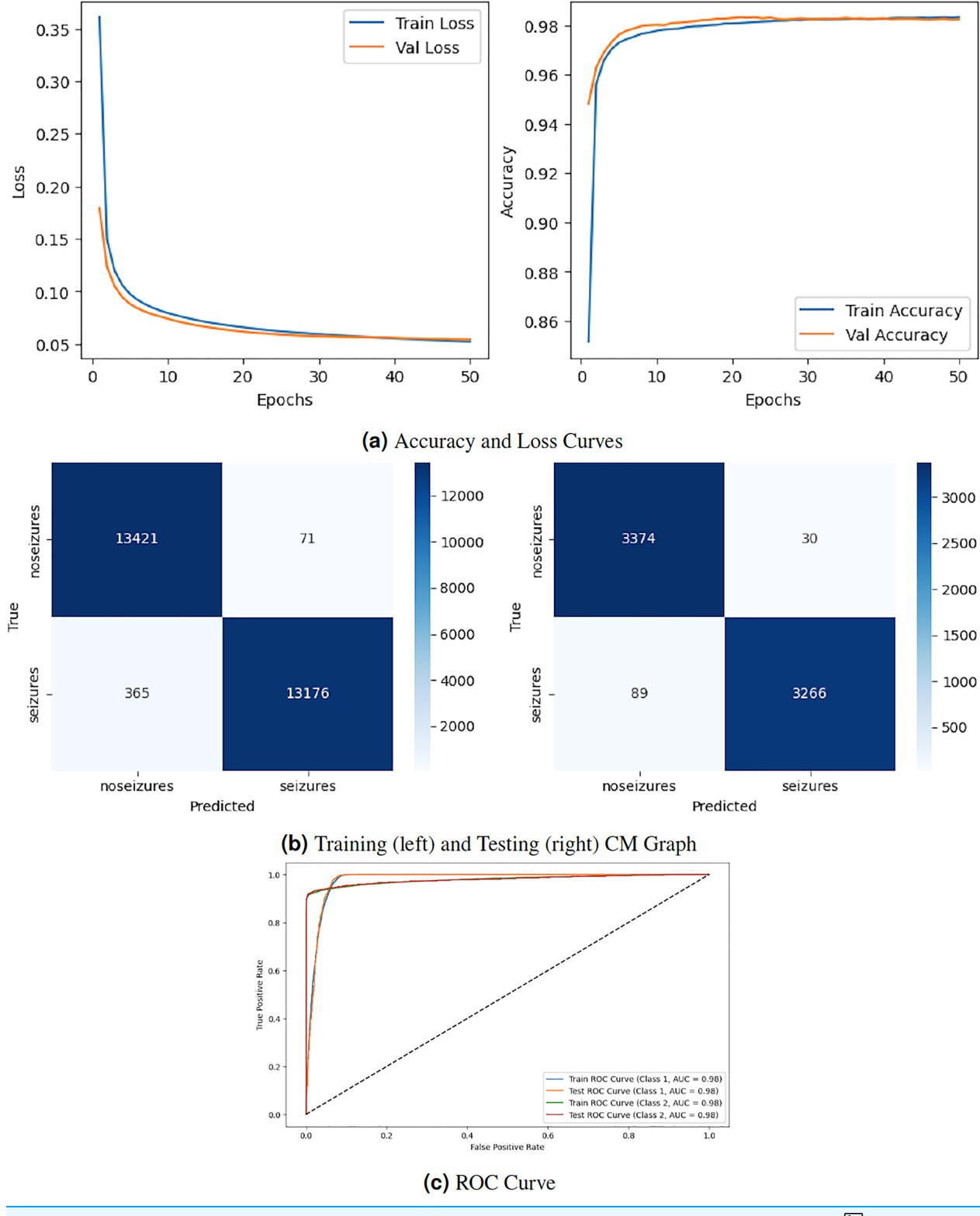

**(a)** Accuracy and Loss Curves

**(b)** Training (left) and Testing (right) CM Graph

**(c)** ROC Curve

**Figure 3  RNN model results.**                               

classes. For Class 1, the train curve area is expressed by the blue line with a 0.98 value, and the orange line displays the test ROC curve with the same value. The train ROC curve for Class 2 is shown by the green line with a value of 0.98, and the red line shows the test ROC curve with the same value for Class 2.

Figure 4 displays the graphical representation of the LSTM model by evaluation training, testing loss, accuracy, CM, and ROC. Figure 4A displays the training and testing accuracy of the model. Training accuracy starts from $0^{th}$ epoch with value 0.940%, and after some fluctuation, it increases to 0.972% at $50^{th}$ epoch. Testing accuracy starts from $0^{th}$ epoch with value 0.943%, and after some fluctuation of increase and decrease, it increases to 0.975% at $50^{th}$ epoch. Figure 4B shows the training and testing loss, which decreases as the number of epochs increases. Training loss starts from $0^{th}$ epoch with value 0.18%, and after some fluctuation increase and decrease, it decreases to 0.09% at $50^{th}$ epoch. Testing loss starts from $0^{th}$ epoch with value 0.17%, and after some fluctuation increase and decrease, it decreases to 0.08% at $50^{th}$ epoch. Figure 4C is a training confusion matrix that performs better because of true positive (13,415) and negative (12,902) values and fewer false positive (77) and negative (639) results. The graph in Fig. 4D is a testing confusion matrix that performs better because of true positive (3,378) and negative (3,221) values and fewer false positive (26) and negative (134) results. Figure 4E displays the ROC curves for both classes in which the train ROC curve for classes 1 and 2 is 0.99, and the test ROC curve for Classes 1 and 2 is 0.99.

Figure 5 displays the graphical representation of the GNN model by evaluation training, testing loss, accuracy, CM, and ROC. The graph in Fig. 5A displays the training and testing accuracy of the model. Training accuracy starts from $0^{th}$ epoch with value 0.955%, increasing to 0.978% at $10^{th}$ epoch. After some minor fluctuation of increase and decrease, training accuracy stops at 0.985% at $50^{th}$ epoch. Testing accuracy starts from $0^{th}$ epoch with value 0.956% increasing to 0.981% at $10^{th}$ epoch. After some minor fluctuation of increase and decrease, training accuracy stops at 0.983% at $50^{th}$ epoch. The graph in Fig. 5B shows the training and testing loss. Training loss starts from $0^{th}$ epoch with value 0.16%, decreasing to 0.05% at $50^{th}$ epoch. Testing loss starts from $0^{th}$ epoch with value 0.16%, decreasing to 0.06% at $50^{th}$ epoch. The graph in Fig. 5C is a training confusion matrix graph that performs better because the proposed strategy produces more continuous, better true positive (10,912) and negative (9,600) values and fewer false positive (4) and negative (1,381) results. The graph in Fig. 5D is a testing confusion matrix graph that performs better because the proposed strategy produces more continuous, better true positive (3,401) and negative (2,965) values and fewer false positive (3) and negative (390) results. The graph in Fig. 5E displays the ROC curves for both classes. For Class 1 and 2, the train and test ROC curve value is 0.98.

## Comparison with conventional ML approaches

Table 4 compares different models for epileptic seizure prediction, focusing on the authors, classifiers used, dataset employed, and the corresponding performance. *Wang et al. (2019)* used the random forest (RF) classifier and achieved a performance of 84.00%.

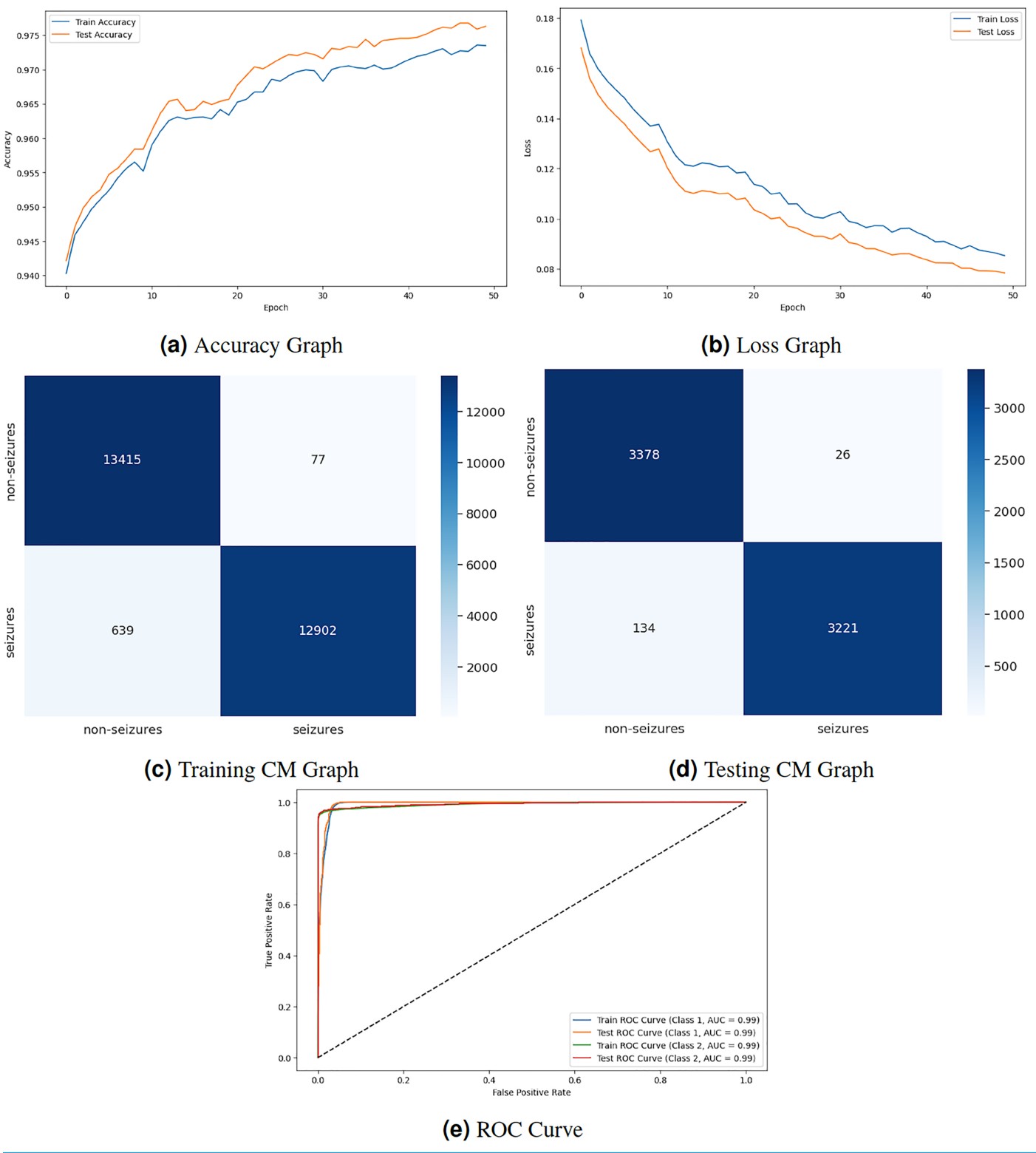

**(a)** Accuracy Graph

**(b)** Loss Graph

**(c)** Training CM Graph

**(d)** Testing CM Graph

**(e)** ROC Curve

**Figure 4  LSTM model results.**

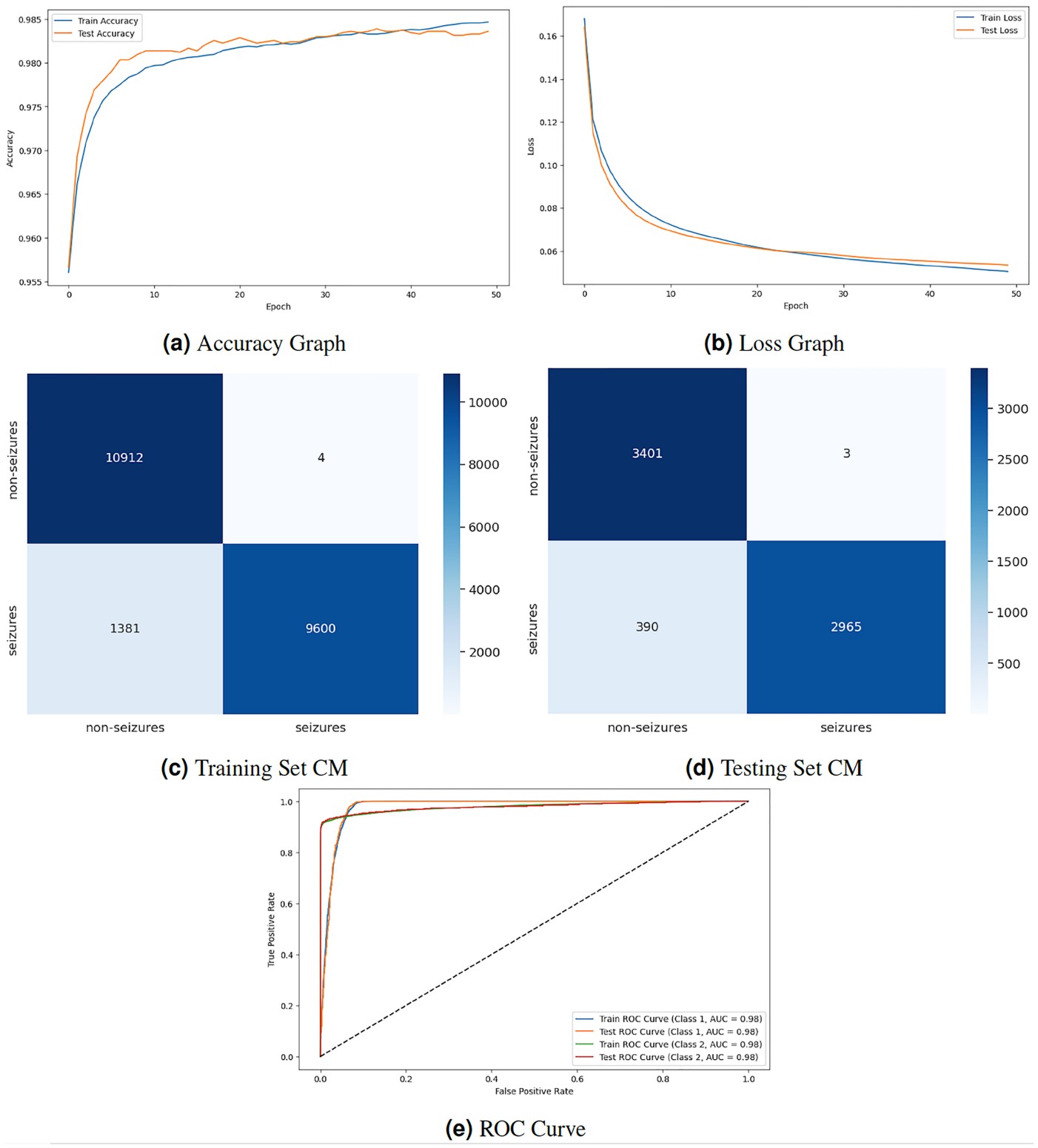

**(a)** Accuracy Graph

**(b)** Loss Graph

**(c)** Training Set CM

**(d)** Testing Set CM

**(e)** ROC Curve

**Figure 5  GNN model results.**

**Table 4 Comparison of the proposed GNN model with existing work.**

| Authors | Classifiers | Dataset | Accuracy |
|---|---|---|---|
| *Wang et al. (2019)* | RF | CHB-MIT | 84.00% |
| *Varnosfaderani et al. (2021)* | LSTM | CHB-MIT | 85.1% |
| *Rasheed et al. (2021)* | DCGAN | CHB-MIT | 88.21% |
| *Dissanayake et al. (2020)* | CNN | CHB-MIT | 88.81% |
| *Zhang, Liu & Chen (2022)* | Transformer | CHB-MIT | 89.50% |
| Proposed model | GNN | CHB-MIT | 98.90% |

*Varnosfaderani et al. (2021)* utilized a two-layer LSTM using the Swish activation function classifier and achieved a performance of 85.1%. *Rasheed et al. (2021)* implemented a deep convolutional generative adversarial network (DCGAN) model for classification and achieved a performance of 88.21%. *Dissanayake et al. (2020)* employed one of the CNN architectures and achieved a performance of 88.81%. *Yu et al. (2022)* utilized a machine learning long short-term memory (MLSTM) model for epileptic seizure prediction and achieved a performance of 89.47%. *Zhang, Liu & Chen (2022)* used a Transformer-based model and achieved a performance of 89.50%. All models, including the proposed model, used the CHB-MIT dataset for epileptic seizure prediction. This research utilizes a GNN as a proposed model and achieves the highest performance with 98.90%. The proposed GNN model outperforms the other models in terms of predictive accuracy, achieving the highest performance. The performance differences between the models are relatively small, but even marginal improvements can be significant in medical applications.

## Discussion and findings

Conventional deep learning methods, such as CNNs, require graph data to be transformed into fixed-size feature vectors, often leading to information loss and disregarding the relational structure. GNNs operate directly on graph structures, enabling them to capture relational dependencies and extract node embeddings that preserve local and global graph properties. Conventional methods need mechanisms for effectively aggregating information from neighbouring nodes in a graph, limiting their ability to exploit graph topology. GNNs leverage message-passing algorithms to propagate and aggregate information across graph nodes, enabling them to capture complex relationships and dependencies. Conventional methods struggle to handle variable graphs and often require fixed-size input representations, making them less adaptable to diverse graph structures.

GNNs are inherently flexible and can operate on graphs of varying sizes and structures, making them suitable for a wide range of graph-based tasks without requiring extensive preprocessing. Conventional methods like image classification may outperform GNNs in certain tasks with well-defined input-output mappings and sample-labelled data. GNNs excel in tasks involving graph data, including node classification, link prediction, and graph classification, where capturing relational dependencies is crucial for accurate predictions. GNNs offer several advantages over conventional DL methods for graph data analysis. By operating directly on graph structures and leveraging message-passing

mechanisms, GNNs can effectively capture relational dependencies and extract valuable insights from complex graph data.

While this approach has demonstrated promising results, we recognize that using heterogeneous GNNs for epileptic seizure detection does not introduce novel improvements or new variant algorithms specifically tailored for this domain. This is an important consideration, as similar methodologies have been extensively researched and applied in other fields. Therefore, future research endeavours should focus on enhancing graph Neural Network models for epileptic seizure prediction by proposing innovative improvements or introducing new variant algorithms. This could include designing custom GNN layers tailored for EEG data or developing novel graph encoding techniques that better capture the unique characteristics of EEG signals.

Moreover, comprehensive comparative studies with existing heterogeneous GNN approaches across different domains should be conducted to highlight the unique benefits and potential adaptations for epileptic seizure detection. Empirical evidence, including ablation studies and evaluations using diverse performance metrics, is crucial to substantiate claims of improved performance or novelty. Future research should also explore novel applications of EEG data in seizure therapy and treatment detection. By addressing technical challenges, fostering interdisciplinary collaboration, and prioritizing patient-centred outcomes, researchers can contribute to developing more effective and personalized approaches to epilepsy management. By acknowledging the current limitations and focusing on these areas for future research, we aim to make substantial contributions to the field, advancing the application of GNNs in medical data analysis and improving the detection and understanding of epileptic seizures from EEG data.

We acknowledge the inherent subject dependency in EEG signals, characterized by non-linearity and variability. Although we did not implement Leave-One-Subject-Out Cross-Validation (LOSO CV), we employed a standard train-test split with stratified sampling to ensure class distribution and reduce bias. Feature standardization was applied to normalize the data, and a GNN was utilized to capture complex relationships within the EEG signals. We assessed model performance using various metrics, including accuracy, F1 score, and ROC AUC. Future work will incorporate the LOSO CV for a more robust evaluation of model generalization across subjects.

## Limitation of the proposed approach

Our research demonstrates the efficacy of the proposed GNN model for EEG-based seizure detection, and it is important to acknowledge certain limitations that may affect the generalizability and applicability of our findings. Our model was trained and evaluated exclusively on the CHB-MIT scalp EEG dataset, which, despite being well-established, limits the generalizability of our findings to other datasets with different characteristics. Future studies should validate the model on diverse datasets to ensure broader applicability.

Our evaluation focused on accuracy, precision, recall, and F1-score but did not fully address clinical implications, such as the impact of false positives and false negatives. Incorporating additional metrics like AUC-ROC and analyzing clinical relevance will

provide a more comprehensive evaluation. The complexity of GNNs also poses a challenge to model interpretability, which is crucial for clinical decision-making. Future work should incorporate explainability techniques to enhance transparency and improve trust in clinical applications. While Z-score normalization was effective, exploring a broader range of preprocessing techniques could further optimize performance. Furthermore, the substantial computational resources required for the GNN model may limit its applicability in resource-constrained settings, highlighting the need for optimizing computational efficiency through techniques like model compression, pruning, quantization, or leveraging lightweight GNN frameworks.

## CONCLUSION

Accurate seizure event prediction *via* EEG signal analysis is crucial for the timely administration of appropriate medical interventions. This research presented an EEG-based epilepsy seizure prediction model utilizing RNN, LSTM, and GNN classifiers with the CHB-MIT dataset for classification. The dataset underwent preprocessing steps, including duplicate removal, handling of missing values, signal segment, resampling, label encoding, and normalization. The proposed GNN model achieved a classification accuracy of 98%, outperforming conventional models such as RNN and LSTM. Additionally, the GNN demonstrated significantly lower computational time, enhancing its feasibility for real-world applications.

The results of this study underscore the capability of the proposed GNN model to effectively predict epileptic seizures with a high level of accuracy and efficiency. Despite its high performance, future research should validate the findings on external datasets and incorporate feature selection techniques to optimize the model further. Moreover, exploring additional deep-learning architectures for seizure detection can provide further insights. The proposed GNN model offers a promising and efficient approach for epileptic seizure prediction, paving the way for enhanced epilepsy management and improved patient care. Continued research and clinical validation are essential to confirm its effectiveness in diverse real-world settings.

### Funding

This work was supported by the Deanship of Research and Graduate Studies at King Khalid University, small Research Project under grant number RGP1/321/45. The funders had no role in study design, data collection and analysis, decision to publish, or preparation of the manuscript.

### Grant Disclosures

The following grant information was disclosed by the authors:
Deanship of Research and Graduate Studies at King Khalid University, small Research Project: RGP1/321/45.

## Competing Interests

The authors declare that they have no competing interests.

## Author Contributions

- Areej Alasiry conceived and designed the experiments, performed the experiments, analyzed the data, performed the computation work, prepared figures and/or tables, authored or reviewed drafts of the article, and approved the final draft.
- Gabriel Avelino Sampedro conceived and designed the experiments, performed the experiments, analyzed the data, performed the computation work, prepared figures and/or tables, authored or reviewed drafts of the article, and approved the final draft.
- Ahmad Almadhor conceived and designed the experiments, performed the experiments, analyzed the data, prepared figures and/or tables, authored or reviewed drafts of the article, and approved the final draft.
- Roben A. Juanatas conceived and designed the experiments, prepared figures and/or tables, authored or reviewed drafts of the article, and approved the final draft.
- Shtwai Alsubai conceived and designed the experiments, prepared figures and/or tables, authored or reviewed drafts of the article, and approved the final draft.
- Vincent Karovic conceived and designed the experiments, prepared figures and/or tables, authored or reviewed drafts of the article, and approved the final draft.

## Data Availability

The raw dataset is available in the Supplemental File.

## Supplemental Information

Supplemental information for this article can be found online at http://dx.doi.org/10.7717/peerj-cs.2765#supplemental-information.

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
