# Peer review of "Epileptic seizures diagnosis and prognosis from EEG signals using heterogeneous graph neural network"

_PeerJ Computer Science, doi:10.7717/peerj-cs.2765_

## Round 0.1 · original submission · Major Revisions

Dear authors,
You are advised to critically respond to all comments point by point when preparing an updated version of the manuscript and while preparing for the rebuttal letter. Please address all comments/suggestions provided by reviewers, considering that these should be added to the new version of the manuscript.

Kind regards,
PCoelho

Reviewer 1 ·

Basic reporting

This paper introduces a novel method that utilizes a heterogeneous graph representation-based Graph Neural Network (GNN) model to detect epileptic seizures from EEG data. The approach uses the CHB-MIT EEG dataset, a well-known resource in epilepsy research, for training and evaluation. The method includes preprocessing steps such as label encoding, normalization, and thorough data analysis, followed by developing and assessing the GNN model. It is compared with other models like Long Short-Term Memory (LSTM) and Recurrent Neural Network (RNN). Experimental results show that the proposed GNN model outperforms traditional deep learning algorithms, achieving an impressive accuracy of 89.0%, thus demonstrating its effectiveness in predicting epileptic seizures.

Experimental design

no comment

Validity of the findings

1. The recall for the Seizures class is relatively low (86.00%) compared to LSTM (91.00%). This limitation suggests that the GNN may fail to detect some seizure events, leading to potential risks in real-world applications. Please clarify this.
2. Although the GNN outperforms RNN and LSTM in terms of overall accuracy and F1-score, the performance gain is incremental (approximately 2%). It may not justify the additional computational complexity or training effort required for GNN models.
3. GNN models are computationally intensive compared to RNN and LSTM. Depending on the dataset size and deployment requirements, their usability in real-time or resource-constrained environments might be limited. Complexity analysis is required.

Additional comments

After reading this work, I realized it is intriguing and applicable to various tasks. However, to make this work more complete, I have the following suggestions:
1. The contribution of this work is not clearly defined. The authors claim that doctors who analyze large amounts of neuroimaging data often end up with inaccurate diagnoses and prognoses of epileptic seizures. Eye strain from evaluating various functional or structural imaging modalities can contribute to this. However, the results from the proposed GNN are not accurate enough to address this issue. Please provide more details on this aspect of the work.
2. Identify the research gap in your work. There are numerous studies on epileptic seizure identification. The authors should specify how their work differs from other studies.
3. The research uses only one dataset, which affects the conclusion that epileptic seizure identification using the proposed GNN model cannot yet be definitively determined. Tests with at least one or two additional datasets are recommended to draw more conclusive results.
4. Check the format of the manuscript, including the citation style.

·

Basic reporting

1. Please improve the figure quality, use minimum 300 dpi (Figure 1, Figure 3, 4 and 5 are hard to read)
2. Consider Mention the objective result in the conclusion

Experimental design

3. The CHB-MIT dataset consists of 22 patients, including males and females, with numerous recordings comprising different montages. How did the authors deal with the different montages of the CHB-MIT dataset? For example, the montage FC1-Ref exists in patients CHB11 and CHB12 but not in patient CHB15. Please add a brief discussion regarding this issue
4. Please add more information about how the data is prepared. Did the author cut the signal to be processed? Did the data has the same length?
5. Addressing Subject Dependency in EEG Signals: EEG signals have properties such as being non-linear, non-stationary, and subject-dependent. Please address how the proposed process deals with the subject dependency issue. Did the authors consider using inter- or intra-subject testing?

Validity of the findings

6. It would be beneficial if the authors clearly present the study's findings in the abstract, including the computational results. Highlight Why is it necessary to use a new perspective to analyze EEG signals? Are other methods considered insufficient? Provide a clear and detailed explanation regarding this matter.
7. What does the author means by traditional approach? Is it machine learning based method? I don't think it is proper to mentioned it as traditional

---

## Round 0.2 · accepted · Accept

Dear authors, we are pleased to verify that you meet the reviewer's valuable feedback to improve your research.

Thank you for considering PeerJ Computer Science and submitting your work.

Kind regards
PCoelho

Reviewer 1 ·

Basic reporting

no comment

Experimental design

no comment

Validity of the findings

no comment

Additional comments

In the revised version of this article, the researcher has made all the necessary adjustments according to the provided recommendations. The content of the article is complete and appropriate for publication.